# Glenoid Inclination: Choosing the Transverse Axis Is Critical—A 3D Automated versus Manually Measured Study

**DOI:** 10.3390/jcm11206050

**Published:** 2022-10-13

**Authors:** Marc-Olivier Gauci, Adrien Jacquot, François Boux de Casson, Pierric Deransart, Hoël Letissier, Julien Berhouet

**Affiliations:** 1Institut Universitaire Locomoteur et du Sport, CHU de Nice, 06000 Nice, France; 2Unité de Recherche Clinique Côte d’Azur (UR2CA), Université Côte d’Azur, 06000 Nice, France; 3Center of Joint and Sports Surgery (ARTICS), 54270 Essey-lès-Nancy, France; 4Tornier, 38330 Montbonnot-Saint-Martin, France; 5SDOD, 38410 Saint-Martin d’Uriage, France; 6Department of Orthopedic Surgery, Hôpital de la Cavale Blanche, 29200 Brest, France; 7Department of Orthopedic Surgery, CHRU de Tours, 37000 Tours, France

**Keywords:** glenoid inclination, 3D shoulder modelization, preoperative planning, transverse axis, trigonum scapulae

## Abstract

The aim of this study was to evaluate the variation in measured glenoid inclination measurements between each of the most used methods for measuring the scapular transverse axis with computed tomography (CT) scans, and to investigate the underlying causes that explain the differences. Methods: The glenoid center, trigonum and supraspinatus fossa were identified manually by four expert shoulder surgeons on 82 scapulae CT-scans. The transverse axis was generated either from the identified landmarks (Glenoid-Trigonum line (GT-line), Best-Fit Line Fossa (BFLF)) or by an automatic software (*Y*-axis). An assessment of the interobserver reliability was performed. We compared the measured glenoid inclination when modifying the transverse axis to assess its impact. Results: Glenoid inclination remained stable between 6.3 and 8.5°. The variations occurred significantly when changing the method that determined the transverse axis with a mean biase from −1.7 (BFLF vs. *Y*-axis) to 0.6 (BFLF vs. GT-line). The *Y*-axis method showed higher stability to the inclination variation (*p* = 0.030). 9% of cases presented more than 5° of discrepancies between the methods. The manual methods presented a lower ICC (BFLF = 0.96, GT-line = 0.87) with the widest dispersion. Conclusion: Methods that determine the scapular transverse axis could have a critical impact on the measurement of the glenoid inclination. Despite an overall good concordance, around 10% of cases may provide high discrepancies (≥5°) between the methods with a possible impact on surgeon clinical choice. Trigonum should be used with caution as its anatomy is highly variable and more than two single points provide a better interrater concordance. The *Y*-axis is the most stable referential for the glenoid inclination.

## 1. Introduction

Glenoid inclination was first described by Basmajian et al. as a “glenoid slope” that works against the inferior subluxation of the humerus [1]. Since then, no studies had focused on the inclination measurement until the early 2000’s. At that time, Bokor et al. proposed to position and artificially fix the inclination at 90° (“ideal neutral plane”) in order to standardize the glenoid measurements from CT-scans [2]. Later, Churchill et al. assumed that the inclination was variable and proposed to measure the glenoid inclination related to the transverse axis (line from the trigonum to the glenoid center). Multiple 2D and 3D handmade measurement methods could be found in the literature but provide many variations [3,4,5]. In a recent study, Boileau et al. found a near perfect agreement for version between the studied automated software and the other handmade methods; however, only a good agreement for inclination [6]. Software versions have evolved since then. More recently, Webb et al. assessed the interrater reliability between different preoperative planning programs and manual measurement in shoulder arthroplasty [7]. They demonstrated reduced reliability of the inclination (ICC = 0.705) from the version (ICC = 0.914) with a measure that differed between 5° to 10° in nearly 50% of cases. However, inclination is a critical factor in our daily practice for diagnosis and classification and is highly involved in certain surgical planning and decision [8,9,10]. Several studies also report that implants inclination in reverse shoulder arthroplasty has been proven to have a direct impact on the clinical outcomes, such as prosthetic stability [11,12], or glenohumeral kinetics and patients’ final motions [13,14]. Thus, we assume those differences must be analyzed in the light of various reference systems that generate the scapula’s transverse axis. 

The aim of our study was to answer the following questions: What are the variations of inclination measurements between each of the most used transverse axis? What are the root causes explaining the differences between the inclination measurement methods? 

Our hypothesis was that the glenoid inclination measurement depends on the chosen referential, and especially the transverse axis. 

## 2. Methods

### 2.1. Study Cohort

From our local database, we included 82 CT-scans of patients with normal shoulders (*n* = 26) and suffering from shoulder arthritis (*n* = 56). Normal shoulders were previously obtained in the setting of polytraumatic injury, traumatic head injury, chronic acromioclavicular joint dislocations, or unilateral shoulder trauma with a contralateral normal shoulder. The entire scapulae were included to detect the most inferior and the most medial points. The etiologies for the pathologic glenoid, as well as modified-Walch [15,16] and Favard [17] classifications, are detailed in Table 1.

### 2.2. Data Acquisition

All 82 CT-scan series were acquired in a supine position, elbow at side, with the following protocol: 120 to 140 kV, 240 mA, pitch ≤ 0.9, rotation time ≤ 1 s. and maximum 1.2 mm slice increment. The field of view should include the entire scapula. The scapula was segmented with the Blueprint software (v2.1.6, Tornier SAS, Montbonnot-Saint-Martin, France). This segmentation was previously found to be valid and reliable with a mean error of 0.4 ± 0.09 mm [18,19].

### 2.3. Method Glenoid Inclination Measure

Three input data are necessary to compute the glenoid inclination:The scapula frontal plane.The transverse axis.The glenoid mediolateral axis.

The glenoid inclination was calculated by projecting the two axes (transverse axis and glenoid mediolateral axis) on the scapula frontal plane. Then, the angle between the two projected lines was measured and provided the glenoid inclination. The transverse axis was the only variable in our study. 

### 2.4. Input Data Definitions

Several definitions exist for the transverse axis. Furthermore, the acquisition method for the transverse axis differs depending on its definition (Table 2).

The scapular frontal plane, the *Y*-axis and the glenoid mediolateral axis were automatically computed by the Blueprint software [6,10,18,19,20] (Figure 1).

For the manual input data, four experienced surgeons used a dedicated software to select the following landmarks on the 3D automatically segmented scapula model of each patient (Figure 2):The trigonum: point at the intersection of the scapular spine and the medial border of the scapula.The glenoid center: a point determined by the center of the segment between the upper and lower apex of the glenoid.Five points were picked regularly along the bottom of the supraspinatus fossa and used to define the best-fit line fossa, as described by Terrier et al. [5].

After the placement step, the resulting values of the glenoid inclination were averaged between the four surgeons.

We compared the measured inclination values when modifying the transverse axis.

### 2.5. Statistical Analysis

We performed a prior power analysis (parameters: mean positioning error = 5 mm, std = 10 mm, α = 5% and β = 20%), indicating the minimum number of cases for the cohort was 63. The intraclass coefficient of correlation (ICC) was calculated for each method from the comparison of the inclination obtained by the four observers.

To evaluate the concordance of the glenoid inclination values given by different measurement methods, we used the following method:We calculated the linear correlation coefficient between the two samples.Then, if significant (r > 0.7), we interpreted a Bland and Altman graph and see if the chosen predefined thresholds were met: maximum mean arithmetic bias at 3° and/or 95% confidence interval at 5° [21].If the value was below the established thresholds (mean or confidence interval), the method was declared as concordant and we did not calculate the Lin’s concordance coefficient in that case.If the value was over the established thresholds, to interpret the source of the mismatch, the Lin’s concordance coefficient was calculated, making it possible to assess whether it was a lack of precision (LoP, %) or a lack of accuracy (LoA, %). The sum of both LoP and LoA was 100%.

Statistical analyses were performed using the MedCalc (v19.4.0, MedCalc Software Ltd., Ostend, Belgium).

## 3. Results

### 3.1. Glenoid Inclination Values and ICC

Significant variations occurred when changing the method that determined the transverse axis with a mean inclination value over 8° for the GT line and BFLF methods that was significantly higher than the inclination at 7° obtained by using the *Y*-axis (*p* = 0.001 and *p* < 0.0001, respectively) (Table 3). Similarly, the range of inclination value varied by modifying the transverse axis: the minimum and maximum inclination values were noticed for the GT line and BFLF methods (−19.5° to 27.8° instead of −15° to 21°).

The most important observed difference for an individual case was 14°. A difference of 5° or higher was observed in 18% of cases between the BFLF and GT-line methods, in 12% of cases between the *Y*-axis and BFLF methods and in 13% of cases between the *Y*-axis and the GT-line methods (Figure 3). In those situations, the *Y*-axis method showed a higher stability to the inclination variation. In one case the *Y*-axis method provided an inclination difference at more than 5° from the two others compared to seven cases for the GT-line method and in seven cases for the BFSF method. This difference was significant (*p* = 0.030).

Regarding the ICC, the automatic method (*Y*-axis) was at 1.00 as expected. The manual methods presented a lower ICC that remained over 0.95 for the BFLF method and under 0.90 for the GT-line method with the widest dispersion (Table 4). 

### 3.2. Identification of the Differences between the Inclination Measurement Methods

The inclination was sensitive to the changing in the transverse axis (Table 5, Figure 4). More specifically, when using the GT-line method, we observed an overestimation of 1° compared to the *Y*-axis generated by the Blueprint software, and of 2° when using the BFLF. This systematic difference of the inclination was more constant with the BFLF method (CCC = 0.91), while it was more due to the LoP (=88%) for the GT-line and the positioning of the trigonum. 

## 4. Discussion

Measuring the glenoid inclination requires determining the transverse axis. To date, the presented three-dimensional referential are published and commercialized and are the most used planning software in the shoulder surgeon’s practice [5,6,10,22,23]. It appears that the transverse axis was a critical referential and its choice has an important impact on the measured glenoid inclination. Moreover, as the preoperative planning has become widely used in our practice, substantial variations in the final position of the implants and clinical results can be expected from the variability of the glenoid inclination that is generated by multiple available software with various methods. We assume that surgeons must understand and control this variability for better management of their patients. 

### 4.1. Inclination Variability

Bokor et al. evaluated the variation of glenoid version measurements by scapula rotation and reported that the neutral position showed the least variation in 1999 [2]. However, more recent articles report that glenoid inclination by itself is not fixed. Indeed, it is highly variable in normal shoulders [20,24] and is often altered in arthritic shoulders [25]. Thus, the inclination variability observed by the surgeon in his daily practice may be firstly due to the glenoid interindividual variation or its pathological condition—not depending on the inclination measurement methods—but also to the choice of the measurement referential. Despite the little information provided in the literature regarding the accuracy of various 3D-planning software programs, surgeons must be aware of the measurements that correlate directly to actual scapulae, the correlations to the standard handmade measurements and the algorithms used to produce the calculations [26]. 

Previous published articles reported that, in most cases, inclination measurements did not differ significantly between preoperative planning systems [27]. This was partially confirmed in our study in which the concordance was very good and the mean biases under 2° on average. Denard et al. reported that with inclination measurements the difference between Blueprint (Tornier), that uses the *Y*-axis, and VIP (Arthrex) that uses the GT-line, was less than 5° in 54% of cases. However, this difference reached 5°–10° in 27% of cases, and was greater than 10° in 19% of cases [28]. Thus, one must be careful regarding the interpretation of the global results and the dispersion must be cautiously analyzed. Similarly, in our study, we were concerned about a few “worse cases” (9%) for which the GT-line method (seven cases) or the BFLF method (seven cases) gave more than 5° difference from the two others which generated a similar inclination value. In one case, the difference reached 14°. This discrepancy was found in one unique case with the *Y*-axis method that reflected a better stability of this method. 

Even if it concerns one case over 10 in our practice and clinical situations, we must warn surgeons about such inclination variations as they may use these measurements, provided by software, to decide on the glenoid implant positioning in anatomic or reverse shoulder arthroplasty based on current recommendations. Thus, the degree of preoperative inclination represents an important factor in surgical decision-making as it is strongly associated with postoperative inclination [29]. Moreover, recommendations for anatomic total shoulder arthroplasty are to not position the glenoid implants over 10° of superior inclination and studies of glenoid component malpositioning in TSA have linked excessive superior tilt with a risk of secondary rotator cuff dysfunction and worse outcomes [30,31]. A third point is that patient-specific instrumentation, navigation, or robot will apply blindly the planning established from measurements given by their software and they are more and more used in current practice. Finally, a reliable inclination is mandatory to provide an adapted postoperative inclination of the implants and software designers have the responsibility not to mislead the surgeons they are supposed to assist. 

The reasons behind the differences in software between various companies are often unexplained by the authors [32]; however, a better knowledge of those variables may allow improved templating with subsequent improved accurate implant positioning. That is why, in our article, we tried to understand the implication of the transverse axis as a determining factor of the inclination variability between the current accepted measurement 3D methods. 

### 4.2. Factors Influencing the Transverse Axis 

From our study, we assume that: (1) the choice of a free-hand picking method is submitted to the inter- and intra-rater variability that may exist when the engineer selects the points of interest; (2) a limited number of points makes the result more sensitive to the error of picking compared to an important number of selected points; and finally (3) certain variations in the scapular anatomy—particularly the trigonum—induce several interindividual variations of the transverse axis in some wide proportions on which surgeons must remain cautious.

Regarding the ICC, the inclination measurement was globally very good and manual methods—as expected—provided a lower ICC than automatic methods (with an ICC at 1.00). The best line fossa ICC and its confidence interval remained over 0.90 while the GT-line method was the lowest with a wider dispersion. We assume that this variability may be related to the number of picked points that were used to detect the transverse axis: two for the GT-line and five for the BFLF. Thus, a higher number of landmarks may limit and smooth the discrepancies between the four observers. 

Another issue is that the trigonum positioning and the overall morphology of the global medial side of the scapular blade is highly variable and leads to more difficulties in identifying this medial landmark. Moreover, in the literature, authors give different definitions of the trigonum that can be the most medial point of the scapula [33,34] or the intersection between the spine and the medial border of the scapula [22,26,35,36] (the definition we have chosen in our article). It can also be termed “trigonum scapulae” [34,37,38], “trigonum spinae” [39] or “os trigonum” [40] with no further precision. As a solution to overcome the described limits, some solution, such as ExatechGPS (Exatech, Gainesville, FL, USA) propose to determine the trigonum by an average of three points [7]. All the various descriptions of this anatomic landmark must warn the surgeon about the difficulty to reproduce exactly the published methods using this point and that the inclination may vary in a substantial way for an individual case depending on the engineer that makes the trigonum picking in a preoperative planning process.

Finally, the trigonum is difficult to identify even with a well-established prior anatomic definition. Indeed, Jacquot et al. published the inter- and intra-observer agreements and they found at 0.76 and 0.91, respectively, for the trigonum positioning. The discrepancies occurred mostly along the superoinferior axis [41], while the variation of the glenoid center was lower. Thus, this variability on the trigonum positioning could have a significant impact on inclination (rather than on version) as it mainly occurs on the superoinferior axis. Regarding the extreme inclination values obtained in the present article, it also demonstrates less robustness of the methods that used the trigonum as it is related to certain anatomic variations of the medial scapula aspect. Our study confirms this high variability and highlights its potential clinical impact on the transverse axis determination and the glenoid inclination measure.

### 4.3. Strengths and Limits 

Our study is not without limits. Our population was limited at 82 patients, and we assume that it does not represent the exhaustive situations that may occur in a current clinical practice. Consequently, we were not able to provide a more specific analysis of the normal or pathologic shoulders that could also have an influence on the inclination measure. Thus, our pathological group did not include a sufficient number of glenoid wear severe enough to result in a medialization and/or superoinferior deviation of the glenoid center. The strengths of this study were that the picking work has been performed by four experienced shoulder surgeons who each picked eight points on the 82 scapulae (656 points), with an accurate definition of each step, in order to limit the measurement biases. This allowed us to identify specific cases with abnormal differences in inclination between the three methods, however more inclusion and in-depth selection criteria are needed to achieve a specific analysis of this particular cases. 

## 5. Conclusions

Methods that determine the scapular transverse axis could have a critical impact on the measurement of the glenoid inclination. The *Y*-axis, GT-line and BFLF methods to generate the transverse axis provide an overall very good concordance in inclination measurement (<2° difference on average). However, a non-negligeable number of cases (around 10%) may provide high discrepancies (≥5°) between the studied methods with possible impact on surgeon clinical choice. We assume that a referential line determined from multiple points (BFLF method or *Y*-axis) provide a better interrater concordance than two single points (GT-line method). Trigonum should be used with caution as it is highly variable as well as the medial edge of the scapula. The *Y*-axis method provides the most stable referential to calculate the glenoid inclination. 

## Figures and Tables

**Figure 1 jcm-11-06050-f001:**
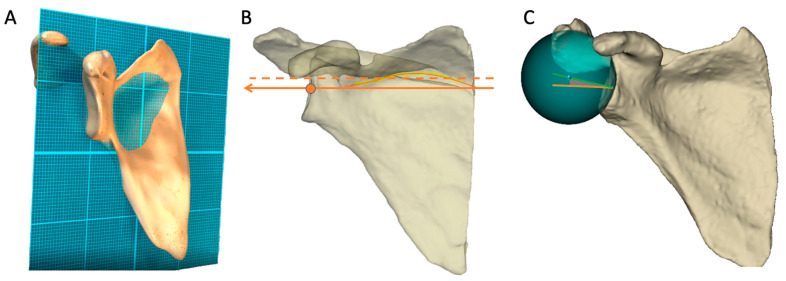
After an automatic segmentation, the following referential were automatically generated from the Blueprint software. (**A**) The plane of the scapular body was generated as the mean plan of all the points of the scapula except the glenoid, the acromion and the coracoid. (**B**) The *Y*-axis was determined by all the points at the intersection of the scapular body and the scapular spine (orange dotted line) and translated through the center of the glenoid (orange arrow). (**C**) The glenoid best fit sphere was the sphere that best fit to the automatically identified glenoid surface. The inclination is measured as angle between the transverse axis (orange line) and the line (green line) between the glenoid center and the best-fit sphere center (blue point).

**Figure 2 jcm-11-06050-f002:**
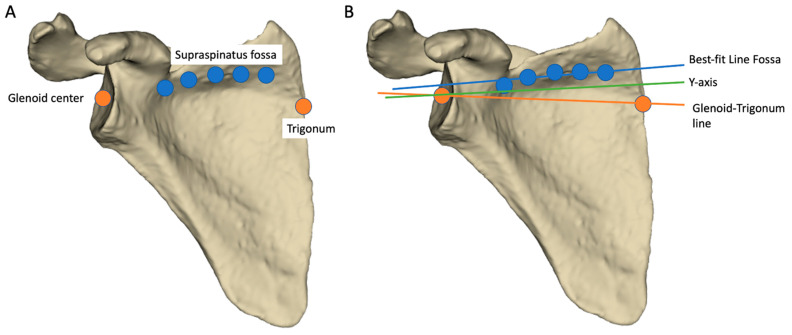
Free-handed landmark picking on a 3D scapular model to determine the transverse axis. (**A**) Each of the four surgeons identified the glenoid center, the trigonum and the supraspinatus fossae on the 82 scapulae. (**B**) The transverse axis varied according to its definition: Best-fit line Fossa was generated from five regular landmarks positioned at the bottom of the supraspinatus fossa, *Y*-axis was automatically generated from all points at the intersection of the scapular spine and the scapular body, Glenoid-trigonum line joined the glenoid center and the trigonum.

**Figure 3 jcm-11-06050-f003:**
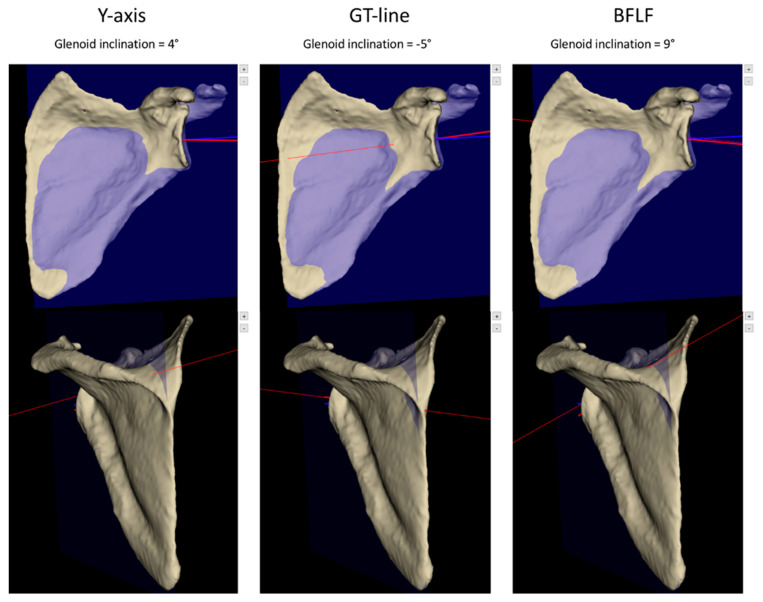
Example of a highly discordant case between the three methods to determinate the transverse axis with an inclination varying from −5° to 9°. The blue line represents the glenoid centerline and the red line represents the transverse axis. GT-line: Glenoid-Trigonum line; BFLF: Best Fit Line Fossa.

**Figure 4 jcm-11-06050-f004:**
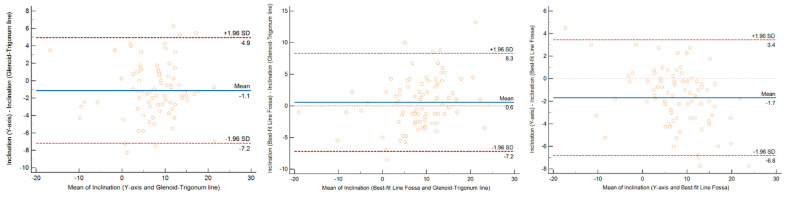
Analysis with the Bland and Altman method of concordance between the measurements of glenoid inclination using the three methods. The widest limits of agreement were found between the Best Fit Line Fossa method and the Glenoid-Trigonum line. SD = Standard Deviation.

**Table 1 jcm-11-06050-t001:** Population.

Age (years)	61 ± 17
Men/Women	26/56
Diagnosis	
Normal shoulder	26
PGHOA	
Type A	11
Type B	18
Type C	1
Type D	1
Massive Rotator Cuff Tear (E0)	5
Cuff Tear Arthropathy	
Type E1	5
Type E2	2
Type E3	4
Post traumatic arthritis	4
Rhumatoid arthritis	2
Post-instability	2
Avascular osteonecrosis	1
Total	82

PGHOA: Primary glenohumeral osteoarthritis.

**Table 2 jcm-11-06050-t002:** Three methods of transverse axis definition used to compute the glenoid inclination.

Input Data Name	Definition	Acquisition Method
Transverse axis		
*Y*-axis [20]	Best-fit line to all points at the intersection of the scapular spine and the scapular body	Fully Automated
Glenoid-Trigonum line [3]	Line through the glenoid center and the trigonum	Need a manual picking
Best Line Fossa [5]	Best-fit line to five regular landmarks positioned at the bottom of the supraspinatus fossa	Need a manual picking

**Table 3 jcm-11-06050-t003:** Inclination values according to the used methods.

	Transverse Axis		
	*Y*-Axis	Glenoid-Trigonum Line	Best-Fit Line Fossa
Mean (°)	6.8	8.0	8.5
Minimum (°)	−15.0	−18.5	−19.5
Maximum (°)	21.0	25.0	27.8
Standard Deviation (°)	6.8	6.6	7.8

**Table 4 jcm-11-06050-t004:** ICC in inclination measures (four observers).

	ICC	95%CI
Y-axis (Blueprint software automatic measure)	1.00	1.00; 1.00
Glenoid-Trigonum line	0.87	0.78; 0.92
Best fit line fossa	0.96	0.94; 0.97

ICC: Intraclass Coefficient of Correlation; CI; Confident Intervals.

**Table 5 jcm-11-06050-t005:** Correlation and concordance between the methods.

Variable Referential	Compared Methods	Linear Coefficient of Correlation		Concordance				
		Pearson (r)		Bland & Altman		Lin		
		r	95%CI	Mean Biase (°)	95%CI (°)	CCC	LoP	LoA
Transverse axis								
	*Y*-axis vs. Glenoid-Trigonum line	0.89	(0.84; 0.93)	−1.1	(−7.2; 4.9)	0.88	88%	12%
	Best-fit Line Fossa vs. Glenoid-Trigonum line	0.86	(0.80; 0.91)	0.6	(−7.1; 8.3)	0.85	89%	11%
	*Y*-axis vs. Best-fit Line Fossa	0.95	(0.92; 0.97)	−1.7	(−6.8; 3.4)	0.91	61%	39%

CI: Confident Intervals; CCC: Concordance Correlation Coefficient; ICC: Intraclass Correlation Coefficient; LoP: Lack of Precision; LoA: Lack of Accuracy.

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
