# Peer review of "Glenoid Inclination: Choosing the Transverse Axis Is Critical—A 3D Automated versus Manually Measured Study"

_jcm, 2022, doi:10.3390/jcm11206050_

Round 1

Reviewer 1 Report

GENERAL COMMENTS

Thank you for your efforts in this work. The authors aimed to evaluate the variation in measured scapular inclination among the three most used methods (Y-axis, Glenoid-Trigonum line, and Best-fit Line Fossa) of measuring the transverse axis using computed tomography (CT) scanning and to investigate the underlying causes that explain these differences. The present study found that the Y-axis method provides the most stable referential to calculate the glenoid inclination and Best-fit Line Fossa, a reference line determined from multiple points, provides better mutual agreement than Glenoid-Trigonum line, which is determined from two single points. The results of this study suggested that the selection of the transverse axis in the software leads to an accurate measurement method for the glenoid inclination angle.

SPECIFIC COMMENTS

Abstract:

Line 18-20: The general phrase "The aim of this study~" may be more appropriate. As a specific example, the aim of this study was to evaluate the variation in measured glenoid inclination measurements between each of the most used methods for measuring the scapular transverse axis with computed tomography (CT) scans and to investigate the underlying causes that explain the differences.

Line 24: Please modify interobservater to interobserver.

Line 28: The "a" is not necessary.

Line 35: Please modify "points" to "points".

Introduction:

Since the Introduction section is very short, it may be worthwhile to explain specifically how the inclination of the glenoid is clinically important. The purpose and hypotheses of this study are mentioned.

Line 46-48: Could you add how specifically determining the precise inclination of the glenoid is clinically important? The "and" should be inserted before the "classification".

Methods:

A prior power analysis has been performed and statistical analysis is described well. Many errors in the order of references were observed in the measurement methods for each transversal axis.

Line 58: Please correct (n=26) and (n=56) as normal shoulder (26) and suffering from shoulder arthritis (56) are confused with the reference.

Table I: Please add the abbreviation PGHOA in the description.

Line 83-84: Please be consistent with the style of the references. Superscript numbers and numbers with () are mixed. Is #9 the correct reference for the Y-axis automatically calculated by the software used in this study? I don't think it is relevant.

Table II:

・Is #10 the correct reference for the Glenoid-Trigonum line? Are the references for this method " Kleim BD, et al. A 3-Dimensional Classification for Degenerative Glenohumeral Arthritis Based on Humeroscapular Alignment. Orthop J Sports Med. 2022 Aug 11;10(8)", " Jacxsens M, etal. A three-dimensional comparative study on the scapulohumeral relationship in normal and osteoarthritic shoulders. J Shoulder Elbow Surg. 2016 Oct;25(10):1607-15", etc.?

・Is #13 the correct reference for the Best-fit Line Fossa? I think the software in #13 uses the Glenoid-Trigonum line. This is an error in reference #4, please correct it.

Line 100: The reference for Terrier et al. is #4. Please correct.

Results:

The evaluation and statistical results of the three methods are described briefly and adequately.

Discussion and Conclusions:

The discussion of the results of this study is well documented.

Line 174: Bokor et al. (17) evaluated the variation of glenoid version measurements by scapula rotation and reported that the neutral position showed the least variation in 1999.

Line 192: A few" is more appropriate than "few".

Line 199-200: Please add a reference regarding the description in “to decide between anatomic and reverse arthroplasty based on cur-199 rent recommendations.”. Does it mean that the glenoid inclination is a selection criterion between TSA and RSA?

Line 211: Please modify “is” to “are”.

References:

Many errors in the order of references were observed in the measurement methods for each transversal axis.

Author Response

Dear Reviewer,

thank you for your comments and propositions. We have followed your recommandations and hope that the manuscript will be more adapted in this last version. 

Regards

The Authors. 

GENERAL COMMENTS

Thank you for your efforts in this work. The authors aimed to evaluate the variation in measured scapular inclination among the three most used methods (Y-axis, Glenoid-Trigonum line, and Best-fit Line Fossa) of measuring the transverse axis using computed tomography (CT) scanning and to investigate the underlying causes that explain these differences. The present study found that the Y-axis method provides the most stable referential to calculate the glenoid inclination and Best-fit Line Fossa, a reference line determined from multiple points, provides better mutual agreement than Glenoid-Trigonum line, which is determined from two single points. The results of this study suggested that the selection of the transverse axis in the software leads to an accurate measurement method for the glenoid inclination angle.

SPECIFIC COMMENTS

Abstract:

Line 18-20: The general phrase "The aim of this study~" may be more appropriate. As a specific example, the aim of this study was to evaluate the variation in measured glenoid inclination measurements between each of the most used methods for measuring the scapular transverse axis with computed tomography (CT) scans and to investigate the underlying causes that explain the differences.

Reply  we modified the sentence as proposed. Thank you

Line 24: Please modify interobservater to interobserver.

Reply  Done. Thank you

Line 28: The "a" is not necessary.

Reply  Done. Thank you

Line 35: Please modify "point" to "points".

Reply  Done. Thank you

Introduction:

Since the Introduction section is very short, it may be worthwhile to explain specifically how the inclination of the glenoid is clinically important.

Reply  Thank you for this comment. We add specific explanation that indicate that inclination is clinically important.

Line 42 to 68: “Glenoid inclination was first described by Basmajian et al. as a “glenoid slope” that works against the inferior subluxation of the humerus[1]. Since then, no studies had focused on the inclination measurement until the early 2000’s. At that time Bokor et al. proposed to position and artificially fix the inclination at 90° (“ideal neutral plane”) in order to standardize the glenoid measurements from CT-scans[2]. Later, Churchill et al. assumed that the inclination was variable and proposed to measure the glenoid inclination related to the transverse axis (line from the trigonum to the glenoid center). Multiple 2D and 3D handmade measurement methods could be found in the literature but provide many variations[3–5]. In a recent study, Boileau et al. found a near perfect agreement for version between the studied automated software and the other handmade methods; however, only a good agreement for inclination[6]. Software versions have evolved since then.

Line 67 to 73: “However, inclination is a critical factor in our daily practice for diagnosis and classification and is highly involved in certain surgical planning and decision[8–10]. Several studies also report that implants inclination in reverse shoulder arthroplasty has been proven to have a direct impact on the clinical outcomes such as prosthetic stability[11,12] or glenohumeral kinetics[13,14]. Thus, we assume those differences must be analyzed in the light of various reference systems that generate the scapula’s transverse axis.

The purpose and hypotheses of this study are mentioned.

Line 46-48: Could you add how specifically determining the precise inclination of the glenoid is clinically important? The "and" should be inserted before the "classification".

Reply  Done. Thank you

Methods:

A prior power analysis has been performed and statistical analysis is described well. Many errors in the order of references were observed in the measurement methods for each transversal axis.

Line 58: Please correct (n=26) and (n=56) as normal shoulder (26) and suffering from shoulder arthritis (56) are confused with the reference.

Reply  Done. Thank you

Table I: Please add the abbreviation PGHOA in the description.

Reply  Done. The description has been added at the bottom of the Table. Thank you

Line 83-84: Please be consistent with the style of the references. Superscript numbers and numbers with () are mixed. Is #9 the correct reference for the Y-axis automatically calculated by the software used in this study? I don't think it is relevant.

Reply  Thank you for this comment. We corrected all the references to make them relative to the text.

Table II:

・Is #10 the correct reference for the Glenoid-Trigonum line? Are the references for this method " Kleim BD, et al. A 3-Dimensional Classification for Degenerative Glenohumeral Arthritis Based on Humeroscapular Alignment. Orthop J Sports Med. 2022 Aug 11;10(8)", " Jacxsens M, etal. A three-dimensional comparative study on the scapulohumeral relationship in normal and osteoarthritic shoulders. J Shoulder Elbow Surg. 2016 Oct;25(10):1607-15", etc.?

・Is #13 the correct reference for the Best-fit Line Fossa? I think the software in #13 uses the Glenoid-Trigonum line. This is an error in reference #4, please correct it.

Reply  Thank you for this comment. We corrected all the references to make them relative to the table.

Line 100: The reference for Terrier et al. is #4. Please correct.

Reply  Thank you for this comment. We corrected all the references to make them relative to the text.

Results:

The evaluation and statistical results of the three methods are described briefly and adequately.

 Reply  Thank you for those comments

Discussion and Conclusions:

The discussion of the results of this study is well documented.

 Reply  Thank you

Line 174: Bokor et al. (17) evaluated the variation of glenoid version measurements by scapula rotation and reported that the neutral position showed the least variation in 1999.

 Reply  We modified our sentence by taking the proposed version that is more relevant.

Line 192: A few" is more appropriate than "few".

 Reply  modified, thank you

Line 199-200: Please add a reference regarding the description in “to decide between anatomic and reverse arthroplasty based on current recommendations.”. Does it mean that the glenoid inclination is a selection criterion between TSA and RSA?

 Reply  Indeed, that is not really exact. We modified the sentence as follow: “as they may use these measurements, provided by software to decide on the glenoid implant positioning in anatomic or reverse arthroplasty based on current recommendations.”

Line 211: Please modify “is” to “are”.

 Reply  modified, thank you

References:

Many errors in the order of references were observed in the measurement methods for each transversal axis.

Reply  all the errors have been addressed. Thank you

Reviewer 2 Report

Line 18 - Please remove "Purpose"

Line 18-20 - Those two sentences are not purposes, but questions. Please re-arrange this part for purposes of this current study

Line 21-22 - This sentence is not grammatically correct. Please re-edit

Line 24 - Please remove a double space before "We"

Line 32 - Please re-arrange the conclusions, including the purpose of the study. Moreover, please improve this sentence The transverse axis is a critical referential to measure glenoid inclination."

Introduction - This part is very poor, without any background about Glenoid Inclination. Second, the reliability part should be developed. And a third part should include a perspective of current and prior studies. Currently, does not provide the reader with a perspective to research.

Method glenoid inclination measure - It should be improved and more specific described

Author Response

Dear Reviewer,

thank you for your comments and propositions. We have followed your recommandations and hope that the manuscript will be more adapted in this last version. 

Regards

The Authors. 

Comments:

Line 18 - Please remove "Purpose"

Reply  Done. Thank you

Line 18-20 - Those two sentences are not purposes, but questions. Please re-arrange this part for purposes of this current study

Reply  Thank you for your comment. We changed for this more adapted sentence: “The aim of this study was to evaluate the variation in measured glenoid inclination measurements between each of the most used methods for measuring the scapular transverse axis with computed tomography (CT) scans and to investigate the underlying causes that explain the differences.”

Line 21-22 - This sentence is not grammatically correct. Please re-edit

Reply  Thank you for your comment. We re-edited this sentence for a more adapted one: “The glenoid center, trigonum and supraspinatus fossa were identified manually by 4 expert shoulder surgeons on 82 scapulae CT-scans.”

Line 24 - Please remove a double space before "We"

Reply  Done. Thank you

Line 32 - Please re-arrange the conclusions, including the purpose of the study. Moreover, please improve this sentence The transverse axis is a critical referential to measure glenoid inclination."

Reply  We modified the sentence as follow to make it clearer: Line “Methods that determine the scapular transverse axis could have a critical impact on the measurement of the glenoid inclination. » Line 32-33.

A similar modification has been done in the conclusion of the manuscript Line 314-315.

Introduction - This part is very poor, without any background about Glenoid Inclination. Second, the reliability part should be developed. And a third part should include a perspective of current and prior studies. Currently, does not provide the reader with a perspective to research.

Reply  Thank you for this comment. we rewritten it to set the context and give the reader more perspectives:

Line 41 to 63: “The glenoid inclination was first described by Basmajian et al. as a “glenoid slope” that works against the inferior subluxation of the humerus2. Since then, no studies had focused on the inclination measurement until the early 2000’s. At that time Bokor et al. proposed to position and artificially fix the inclination at 90° (“ideal neutral plane”) in order to standardize the measurements from CT-scans8. Later, Churchill et al. assumed that the inclination was variable and proposed to measure the glenoid inclination related to the transverse axis (line from the trigonum to the glenoid center). Many 2D and 3D measurement methods could be found in the literature with many variations6,23,26,31. In a recent study, Boileau et al. found a near perfect agreement for version between the studied automated software and the other handmade methods; however, only a good agreement for inclination6. Software versions have evolved since then.”

Line 67 to 73: “However, inclination is a critical factor in our daily practice for diagnosis and classification and is highly involved in certain surgical planning and decision[8–10]. Several studies also report that implants inclination in reverse shoulder arthroplasty has been proven to have a direct impact on the clinical outcomes such as prosthetic stability[11,12] or glenohumeral kinetics[13,14]. Thus, we assume those differences must be analyzed in the light of various reference systems that generate the scapula’s transverse axis.

Round 2

Reviewer 1 Report

Thank you for appropriately revising this paper to the points I pointed out.

The quality of the papers has improved.

I appreciate the effort you have devoted to this work.

Reviewer 2 Report

Thank you